# The pace of hospital life: A mixed methods study

**Janet C. Long**◉*, **Chiara Pomare**◉, **Louise A. Ellis**, **Kate Churruca**, **Jeffrey Braithwaite**◉

Australian Institute of Health Innovation, Macquarie University, Sydney, Australia

* janet.long@mq.edu.au

**Data Availability Statement:** The survey was developed from existing measures of these constructs. We report in detail on individual items: wording, sources (all of which are published) and validation details of those individual items in the

## Abstract

The pace-of-life hypothesis is a socio-psychological theory postulating that citizens of different cities transact the business of life at varying paces, and this pace is associated with a number of population level variables. Here we apply the pace-of-life hypothesis to a hospital context to empirically test the association between pace and patient and staff outcomes. As pressure on hospitals grow and pace increases to keep up with demand, is there empirical evidence of a trade-off between a rapid pace and poorer outcomes? We collected data from four large Australian hospitals, inviting all staff (clinical and non-clinical) to complete a survey, and conducted a series of observations of hospital staff's walking pace and transactional pace. From these data we constructed three measures of pace: staff perception of pace, transactional pace, and walking pace. Outcome measures included: hospital culture, perceived patient safety, and staff well-being outcomes of job satisfaction and burnout. Overall, participants reported experiencing a "fast-paced" "hurried" and "rapid" pace-of-life working in the Australian hospital sector. We found a significant difference in perceived pace across four hospital sites, similar to trends observed for transactional pace. This provides support that the pace-of-life hypothesis may apply to the hospital context. We tested associations between faster perceived pace, hospital culture, staff well-being and patient safety. Results revealed perceived faster pace significantly predicted negative perceptions of organizational culture, greater burnout and lower job satisfaction. However, perceived pace did not predict perceptions of patient safety. Different perceptions of hospital pace-of-life were found between different clinical settings and the type of care delivered; staff working in emergency departments reported significantly "faster-paced" work environments than staff working in palliative, aged care, or rehabilitation wards.

## Introduction

The pace-of-life hypothesis is a socio-psychological theory suggesting that different cities and countries vary in the speed at which their residents experience and interact with the world (i.e., the pace-of-life) [1]. The results of previous studies on the pace-of-life hypothesis have found significant and consistent differences in the pace of routine activities between cities and countries across the globe [e.g., 1–3]. For example, pace was found to be faster in colder

manuscript, sufficient to allow replication. Data cannot be shared publicly because results contain potentially identifiable material. A requirement of our Ethics approval is that data collected remain confidential. Permission to access confidential data (held securely on Macquarie University servers) may be made through application to South Eastern Sydney Local Health District Human Research Ethics Committee (contact via Research Directorate Administration Officer +61 2 9382 3587).

**Funding:** This work is supported in part by NHMRC Investigator Grant APP1176620 and NHMRC Partnership Grant (9100002), chief investigator JB.

**Competing interests:** The authors have declared that no competing interests exist.

climates, economically more productive countries, and in individualistic cultures [1]. Pace-of-life is an important aspect to consider when studying the culture of cities [1].

The pace-of-life hypothesis has recently been theoretically applied to the context of hospitals [4], but to date no research has empirically investigated its potential consequences on variables such as hospital culture and staff wellbeing in this context. The authors hypothesise that there is an optimal pace for hospitals to function well that lies between too fast where mistakes are more likely, staff experience greater burnout, staff retention is compromised and patient satisfaction drops, and too slow where efficiency falls, tasks are not completed, expenses increase, and waitlists get longer.

There is evidence to suggest that the time and activity patterns of health professionals, such as the way nurses distribute their time, the increasingly fragmentary nature of task switching, and the number of interruptions they receive during work activities, play an integral role in shaping organisational culture [4, 5]. Furthermore, organisational research identifies the experience and use of time among staff as an important part of the culture of their organisation which can communicate priorities and approaches to work [6]. Therefore, in the context of this study, pace is thought to provide not only an indication of time management and balancing of work-demands, but a reflection of the organisation's culture [4, 7].

Theories of pace-of-life may hold considerable promise to shed light on the relationship between the hospital environment, patient outcomes and staff well-being. The pace of hospital life may vary depending on context and the type of care delivered (e.g., palliative care may have a slower, gentler pace than intensive care units), however this relationship is yet to be tested empirically [4]. Hospital environments generally are experiencing increasing pressure to accelerate the pace of routine work as a result of demands for services and increasing cost constraints [8]. This increase in demand is associated with faster performance of work [9]. This raises questions around whether this is sustainable [8] and whether a faster pace threatens the quality of care delivered [10].

As hospital staff face pressures to deliver care at a faster pace there is also a potential risk to staff well-being (e.g., burnout, job satisfaction), which can be detrimental to patient safety [11]. Workload and time pressure have been identified as a major contributor to staff burnout [12, 13]. For example, as demand for acute care services increases, hospital staff experience pressure to work at a faster pace to deliver care to patients in need. Working at levels of excessive effort for a period with limited recovery time is a precursor to burnout [12], and burnout is known to affect the quality of care provided to patients [11, 14, 15].

We are the first to apply the theory of pace-of-life and the relationship to culture, patient outcomes and staff well-being to the hospital context. We anticipate the data will allow for insights into hospital organisational culture that are new and important, with potential implications for improving the delivery of patient care in the longer term through interventions targeted at modifying hospital pace. While there may be less conspicuous aspects of organisational cultural context, our study of these constructs reflects a growing recognition that taking care of the "small things" (e.g., time usage, a tidy work space) might make it easier to keep the "big things" in check (e.g., patient safety) [16].

The aim of this study was to empirically test Braithwaite and colleagues' theoretical findings regarding how the pace-of-life hypothesis applies to the hospital context [4] (i.e., is there a difference in pace in different hospitals?); to examine if hospital staff perceptions of pace influence staff well-being, organisational culture, and patient safety; and whether perceptions of pace differ in different clinical settings. We developed and tested the following hypotheses:

1. Hospital pace can be measured in different formats (perception, transaction, walking) consistently.

2. There is a relationship between faster pace and a negative workplace hospital culture.

3. Faster hospital pace is linked to lower job satisfaction.

4. Faster hospital pace is linked to higher burnout.

5. Faster hospital pace is predictive of poor patient safety outcomes.

6. Hospital pace differs based on clinical settings.

## Method

The researcher team employed a convergent, parallel mixed methods design [17]. Data collection involved a survey (to measure perceptions of pace, culture, patient safety and staff well-being), and two modes of structured observation (objective measures of walking and transactional pace). The survey was developed from existing measures of these constructs, details of which are summarised below. All study variables are summarised in Table 1. The ethical conduct of this study was approved by South Eastern Sydney Local Health District (HREC ref no: 16/363). Governance approvals to conduct the research were obtained for each hospital site. Ethics required individual sites to remain anonymous in all reporting.

### Study setting and participants

The study was conducted at four public hospitals in metropolitan Sydney, Australia. The four hospitals were of similar size (all above 500 beds), and varying geographic locations and socio-economic disadvantage across greater metropolitan Sydney [18]. All hospitals offered similar types of services (e.g., emergency department, intensive care, surgical, aged care). All staff (clinical and non-clinical) working at the four hospitals were invited to participate.

For the staff survey, we sought recruitment of 100 respondents from each hospital. Staff were first invited to participate in the survey through an invitation sent to their work email address. The email included a link to a *Qualtrics* [19] online version of the survey. Hard-copy surveys were also made available to staff in tea rooms or common areas. Participation was voluntary and anonymous, with no incentives being offered to enhance enrolment. Consent was assumed through responding to and completing the survey. Demographic data was collected in the survey including age, duration at hospital, hours worked per week, and whether they had direct interaction with patients.

**Table 1. Summary of study variables.**

| | Variables | Source | Details |
|---|---|---|---|
| Pace | Perceived Pace | Survey | Five items from the pace subscale of the Organizational Temporality measure [6]. |
| | Walking Pace | Observation | Timing how long it takes a staff member to walk 20 m (~60 feet) [1]. |
| | Transactional Pace | Observation | Time for the hospital switchboard to answer the phone (from an outside line) and answer a patient enquiry. |
| Culture | Hospital Culture | Survey | Safety Attitudes Questionnaire [25]. Two subscales were combined: teamwork culture (six items) and safety culture (seven items). |
| | Staff Demeanour | Observation | Demeanour of hospital staff (who answered the patient enquiry) was rated based on surliness. |
| Patient Safety | Patient Safety—global | Survey | Single-item measure of an overall patient safety grade [26]. |
| Staff Well-being | Job Satisfaction | Survey | Five items from the Job Diagnostic Survey [27]. |
| | Burnout | Survey | 10-item version of the Maslach Burnout Inventory [29]. |

## Measures of hospital pace-of-life

**Perceptions of pace.** Hospital staff members' perception of pace was assessed using an all staff survey. The pace subscale from Ballard & Seibold's [6] organizational temporality measure was selected. Pace refers to the tempo or rate of activity, as a part of how time is enacted and socially constructed [6, 20–22]. Participants were asked to consider the way they referred to time in the course of carrying out daily tasks at work, and then were presented with five words: "fast-paced", "hurried", "rapid", "quick", "racing". Participants then rated each word on a six-point Likert scale in terms of how strongly they disagreed (1) or agreed (6) with it as a descriptor of their work. A perception of pace score was computed for each respondent by averaging responses across the six items, with higher scores indicating faster, more hurried, rapid, quick and racing pace. Ballard and Seibold (6) reported an acceptable level of internal reliability (0.85). In the present study, our internal consistency coefficient was 0.94.

**Transactional pace.** Transactional pace was measured using observational data. In the original study, Levine and Norenzayan (1) measured the time it took to transact a simple purchase from a post office. Here we used instead, the time to make a routine patient enquiry from the hospital switchboard. Twelve measures were taken from each participating hospital. Four phone calls were made: once in the morning (9:00 to 10:00), once around midday (12:00 to 13:00), once in the afternoon (15:00 to 16:00), and once in the evening (18:00 to 19:00). This procedure was repeated on three days, each one week apart, on a Tuesday, Wednesday, and Thursday. One researcher collected all data points.

Time for the hospital switchboard to answer the phone (from an outside line) was measured as the number of seconds from the start of a ring tone till the call was answered by a person and the patient enquiry was answered. For automated phone answering systems that have a standard length of time till the recording starts, time from being transferred from the recorded system to a person (e.g., if requested to dial 0 to talk to someone) was recorded. The standard question asked, once connected to the operator was "Can you tell me if [name]–that's spelt [––––] was admitted today?" We consistently used the name of one of the authors and provided their date of birth if requested. The timing ended when the response "No, she wasn't admitted" was given and the call terminated. We calculated an average transactional pace score (i.e., the average total time for each hospital to answer the patient enquiry). Outliers were considered based on standard deviation and removed from the dataset.

**Walking pace.** Walking pace was measured using observational data. Following the methodology of Levine and Norenzayan's [1] study, walking speed of staff was measured by timing how long it takes a staff member to walk 20 m (~60 feet). Straight, unobstructed hallways were chosen in each hospital, in public areas (not wards). The software application, WOMBAT (Work Observation Method by Activity Timing) [23], was used to measure the start and end of each walk (a researcher standing at either end of the measured course and signalling start and finish to avoid parallax error). Time was measured to the nearest second. We aimed to collect measurement data on a minimum of 100 separate "walks" at each hospital. Walkers were blinded to the nature of the study and the purpose of the observation, but posters prominently displayed at the sites noted the presence of researchers and the general nature of the data collection activity. Researchers were easily identifiable by their pink shirts with "researcher" printed on them. Staff members were able to opt-out [24] by discretely signalling "no" to the researchers. Staff were included in the data collection and timed if they were: (a) hospital staff recognisable by their scrubs/uniforms or work identification badge, (b) walking alone, and (c) not distracted by speaking on, texting, or looking at a mobile phone screen. Uniforms and security badges were sufficiently different to allow us to identify walkers as nurses, medical officers, domestic staff, allied health or "other" (e.g., security, managers). All walking speeds

were collected between 09:00 and 11:00 in the morning, and 14:00 and 16:00 in the afternoon aiming for a minimum of 30 "walks" per session. We computed an average walking pace score for each hospital.

## Outcome measures

**Culture.** Hospital culture was measured using the Safety Attitudes Questionnaire (SAQ) [25] as part of the survey. We combined two subscales to form a composite measure of culture: teamwork culture (six items) and safety culture (seven items). The SAQ is a validated instrument used to measure attitudes and perceptions in various safety-related domains in healthcare. Example items include: "Nurse input is well received in this clinical area" (teamwork) and "In my work area, it is difficult to discuss errors" (safety). Questions were measured on a five-point Likert-type scale (1 –strongly disagree to 5 –strongly agree), with an option for 'neutral'. Responses to each item were summed to create an overall Hospital Culture score in the range of 0–100. Higher scores indicated a more positive perception of hospital culture. One adaptation was applied to the original items; participants were asked to answer by considering the culture of their "work area", rather than their clinical area. The internal consistency reliabilities of SAQ dimensions were assessed using Cronbach's alpha. In the present study, we found acceptable Cronbach's alpha coefficients for internal consistency (0.90), similar to that reported by Sexton, Helmreich (25) for all sub-scales (0.90).

Culture was also operationalised as 'staff demeanour' and captured in the observational data. When measuring transactional pace, we assessed the demeanour of hospital staff (who answered the question) on a five-point Likert-type scale (1 –surly/unpleasant to 5 – pleasant).

**Patient safety.** Patient safety was assessed with a single-item measure from the Hospital Survey on Patient Safety Culture (HSOPS) [26] included in the staff survey. Specifically, overall grade on patient safety was assessed with a single-item, rated on a five-point Likert scale (1 – excellent to 5 –failing): "Please give this hospital an overall grade on patient safety".

**Staff well-being.** Existing validated scales were selected to measure job satisfaction and burnout in the survey. Job Satisfaction was assessed using five items from the Job Diagnostic Survey (JDS), for example: "In general, I like working here". Responses were rated on a five-point Likert scale (1 –strongly disagree to 5 –strongly agree), with an option for 'neutral'. Scores were summed, with higher scores indicating a greater degree of job satisfaction. Bowling and Hammond [27] reported an acceptable level of internal reliability (0.84). In the present study, we found internal consistency coefficient of 0.87.

Burnout was measured using a 10-item version of the Maslach Burnout Inventory (MBI [28–30]). Due to its appropriateness for use in healthcare settings [31, 32] only two subscales of burnout—emotional exhaustion (five items) and depersonalisation (five items)—were used for the current survey as the third subscale, personal accomplishment, would be less relevant to non-clinical staff. A sample item is "I feel emotionally drained from my work". Items were measured on a seven-point Likert scale (1 –strongly disagree to 7 –strongly agree), with an option for 'neutral'. Scale scores were calculated by averaging items for each subscale, with higher scores indicating higher burnout. The internal consistency of the present study (0.93) was similar to the original (0.91).

**Deepening the exploration of pace in hospitals.** Three open-ended questions were included in the survey. Questions were phrased in a general way to allow participants to reflect on the influence of pace in their work. For example, "Are there other things you notice about the behaviour of people in this hospital that negatively affect peace, cooperation and well-coordinated work?"

## Data analysis

Survey responses and observational pace data were analysed using descriptive statistics followed by regression analyses. All data were analysed via IBM's SPSS Statistics Version 25 [33] and conducted at the 0.05 significance level. Ordinal categorical variables with five or more levels (i.e., Likert responses) were treated as continuous variables (Kleinbaum, Kupper, Nizam, Muller, 2008). Qualitative survey responses (open-response questions) were analysed based on sentiment. Inductive coding was completed for all responses across hospitals, with a specific focus on content related to pace-of-life (e.g., working quickly, feeling time pressured). Data from the different methods of data collection were triangulated to examine the convergence of findings from the different sources [34].

## Results

Pace of hospital life was measured in four hospital sites using three measures: perception of pace, transactional pace, and walking pace. For walking pace, we observed 439 staff (Hospital A: n = 135, Hospital B: n = 106, Hospital C: n = 103, Hospital D: n = 95). Observational data showed it took an average walking pace of 14 seconds (min = 13; max = 14) for hospital staff to traverse 20 meters of a corridor. There was minimal variation in this data, across and within hospitals. The average transactional pace across all four hospital was 40 seconds (min = 16; max = 60) for staff to answer a simple patient enquiry on the main switch phone, with no significant differences across sites, $F(3, 26) = 0.88$, $p = .467$. For perception of pace measured in the survey, hospital staff across all four sites perceived their work as fast-paced, rushed, and time-poor. Free text answers, especially from nurses validated this perception: "*very rushed, running out of time, sometimes there's no time for morning tea break, lunch can be rushed, interruptions, exhausted by mid-day*" [Participant 1, Nurse]. A one-way ANOVA revealed a significant difference in perceptions of pace based on hospital site, $F(3,320) = 3.86$, $p = .010$. Trends of which hospital had a faster pace-of-life were consistent for perceptions of pace and transactional pace, providing partial support for Hypothesis 1. Hospitals are ranked by their pace-of-life in Fig 1. The characteristics of survey respondents are presented in S1 Appendix.

We compared clinical (n = 274; 67.3%) and non-clinical staff (n = 133; 32.7%) and found significant differences based on perceptions of pace ($t(317) = -2.04$, $p = .042$), indicating that clinical staff reported higher scores on the pace scale ($M = 4.13$, $SD = 1.29$) compared to non-clinical staff ($M = 3.80$, $SD = 1.40$) as shown in Fig 2 (i.e., clinical staff perceived their work as more "fast-paced", "hurried", "rapid", "quick" and "racing"). There were no significant differences for clinical vs. non-clinical staff based on culture ($t(337) = -1.10$, $p = .274$), overall patient safety grade ($t(343) = 1.59$, $p = .123$), patient safety issues reported ($t(335) = -1.53$, $p = .126$), job satisfaction ($t(345) = 0.68$, $p = .496$) and burnout ($t(343) = 1.23$, $p = .220$). Hereafter, results are presented in relation to hypotheses.

## Pace and hospital culture

A multiple linear regression examined if staff perceptions of pace predicted organizational culture, considering age, duration at hospital, hours worked per week, and interaction with patients as confounding variables. The model was statistically significant, $F(5, 295) = 3.44$, $p = .005$. The analysis showed that perceived pace ($t(295) = -3.57$, $p < .05$) significantly predicted hospital culture, supporting Hypothesis 2. That is, perceiving daily work tasks in a hospital as more "fast-paced", "racing", "quick", significantly predicted a negative organizational culture. Quantitative results were supported with responses to open-ended questions. Participants suggested that, in the context of a "fast-paced" working environment, there are increases in work pressure and environmental constraints which negatively impact on teamwork culture: "*When*

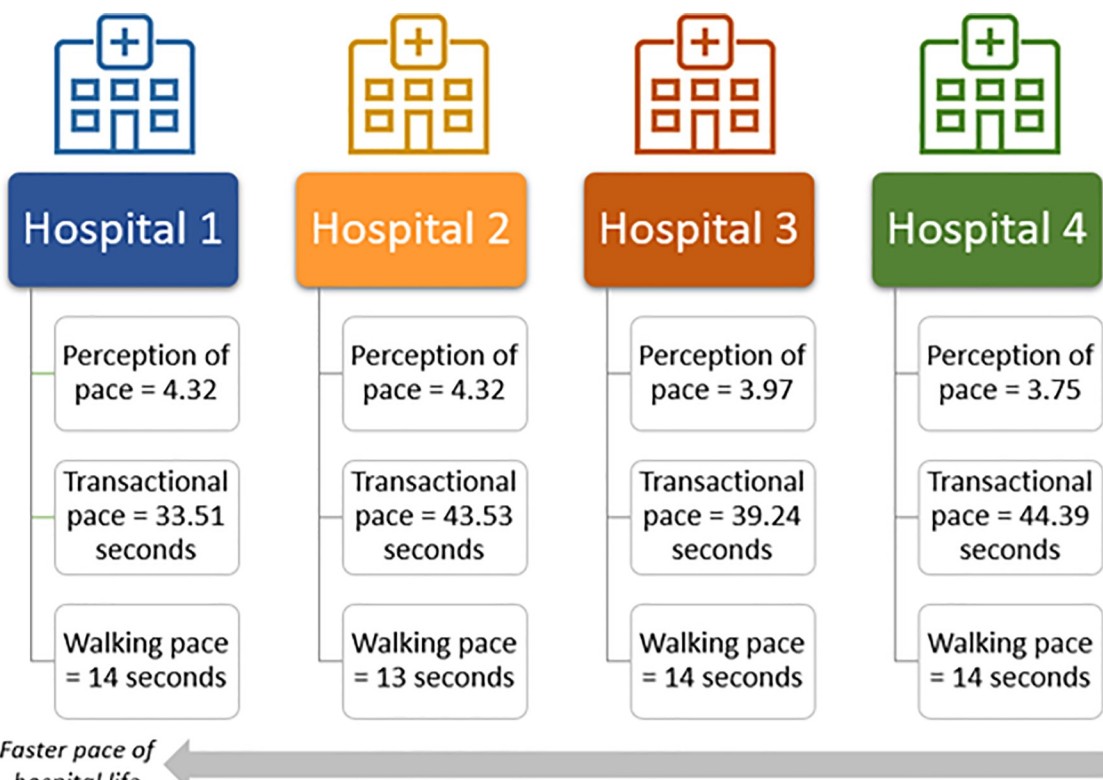

**Fig 1. Hospital sites ranked by their pace of hospital life.**

*we are overwhelmed, especially with no sick leave, annual leave or maternity leave, we all work under strain and can be "bitchy"* [Participant 2, Nurse]; "*Peace, good cooperation and coordination can only happen if there is sufficient staff given sufficient time*" [Participant 3, Pharmacist].

Further, when measuring transactional pace, we also rated the demeanour of hospital staff that answered the main switch phone as an additional measure of culture. We investigated whether the demeanour of hospital staff was a predictor of transactional pace, keeping time of day (morning, midday, afternoon) constant. Demeanour and transactional pace were not significantly related ($p = .278$) and transactional pace did not predict demeanour, while holding time of day constant ($p = .545$).

## Pace and job satisfaction

A multiple linear regression was run to predict job satisfaction from perceived pace, considering age, duration at hospital, hours worked per week, and interaction with patients as confounding variables. The model was statistically significant, $F(5, 301) = 3.35$, $p = .006$. The analysis showed that perceived pace ($t(301) = -4.07$, $p < .05$) significantly predicted job satisfaction. That is, perceiving daily tasks at work in a hospital as *more* "fast-paced" and "quick", significantly predicted low scores of job satisfaction. Thus, our findings support Hypothesis 3.

## Pace and burnout

A multiple linear regression was run to examine whether perceived pace predicted burnout in hospital staff, treating age, duration at hospital, hours worked per week, and interaction with patients as confounding variables. The model was statistically significant, $F(5, 303) = 7.75$, $p <$

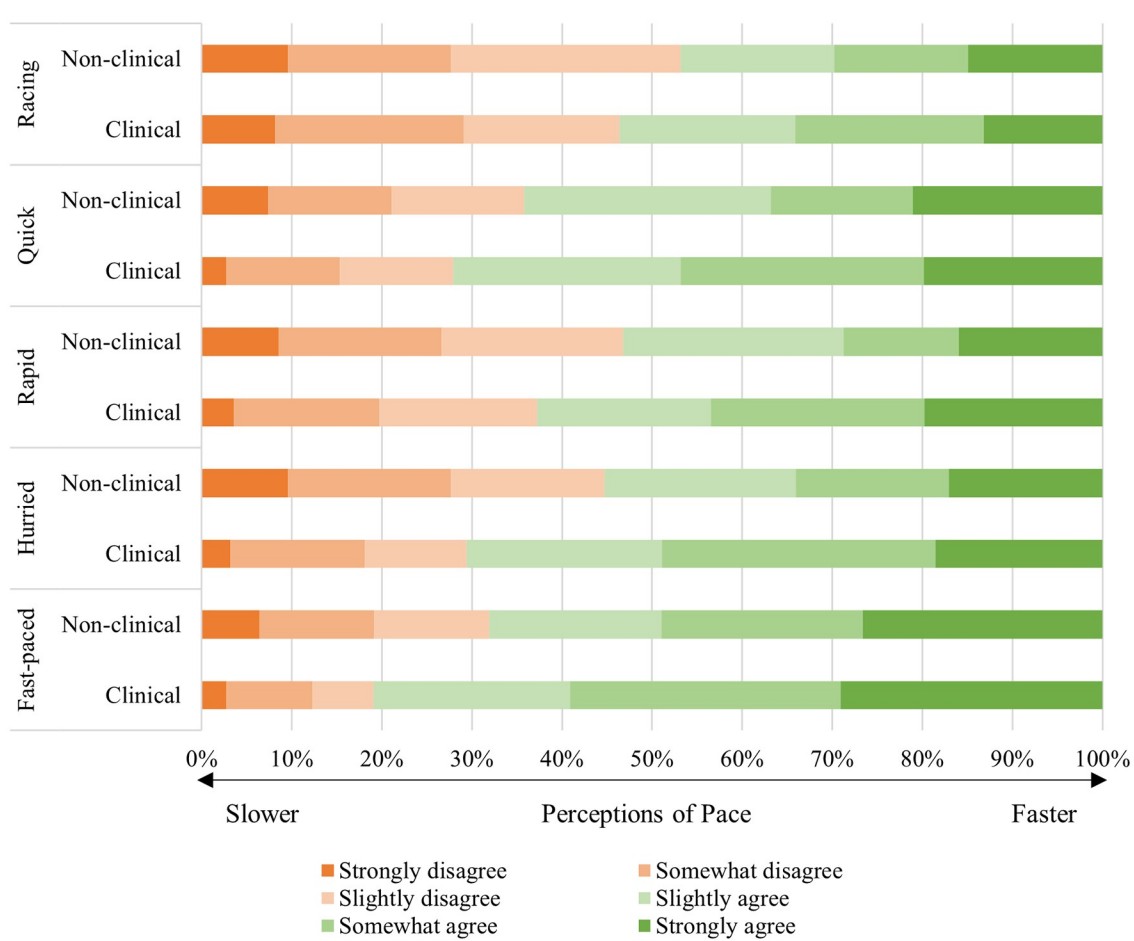

**Fig 2. Perceptions of hospital pace: Clinical vs. non-clinical staff.**

.05. Scoring higher on the pace ($t$ (303) = 5.82, $p$ < .05) significantly predicted burnout. Thus, in support of Hypothesis 4, we found that perceiving daily tasks at work in a hospital as *more* "fast-paced", significantly predicted higher levels of burnout in hospital staff. For some staff, "fast-paced" patient care resulted in issues that increased workload and potentially, burnout: "*Staff, both from my department and others, are focused on how to transfer/discharge the patients in the quickest way possible. This means that issues that should have been addressed are left for the next person to have to deal with*" [Participant 4, Nurse 2].

## Pace and patient safety

We tested whether perceived pace of hospital life predicted patient safety, treating age, duration at hospital, hours worked per week, and interaction with patients as confounding variables. The model was not statistically significant, $F(5, 301) = 1.60$, $p = .160$. Therefore, our findings do not support Hypothesis 5. Responses to open-ended questions suggested that a relationship may still exist between pace of hospital life and safety of patient care. For some staff, the "fast-paced" nature of the hospital brought on a sense of pressure that could lead to negative ramifications for the safety of care delivery: "*Pressure to admit and discharge inappropriately and quickly*" [Participant 5, Doctor].

### Pace in different clinical settings

To examine differences in staff perceptions of pace based on clinical settings, we conducted a one-way ANCOVA to examine if differences in pace existed dependent on the type of care delivered, keeping age, duration at hospital, hours worked per week, and interaction with patients constant. As theorised in previous work [4], we examined if there was a difference in perceptions of pace for staff working in a presumably faster-paced ward (emergency department; $n = 42$) compared to presumably slower-paced wards (palliative care, aged care, and rehabilitation wards; $n = 32$). Across the four hospitals, analysis revealed that staff working in the emergency department perceived their work as significantly more "fast-paced", compared to staff working in palliative care, aged care, and rehabilitation wards, $F(1, 74) = 14.27, p < .05$. This provides support for Hypothesis 6. While staff working in the emergency department had lower mean scores of job satisfaction ($M = 3.32; SD = 0.54$) and higher mean scores of burnout ($M = 3.83; SD = 1.46$) compared to staff working in palliative care, aged care, and rehabilitation wards (Job satisfaction: $M = 3.48, SD = 0.40$; Burnout $M = 3.46, SD = 1.42$), there were no significant differences in staff well-being (job satisfaction or burnout) based on clinical setting.

## Discussion

The study applied the pace-of-life hypothesis to hospitals, examining if pace in a hospital is related to organisational culture, staff well-being, and the safety of patient care; and whether pace is different in different clinical settings. There were differences in the perceived pace of hospital life for different hospitals. Faster perceived workplace pace significantly predicted negative organizational culture, higher burnout, and lower job satisfaction; however, pace did not significantly predict patient safety. Lastly, perceptions of pace significantly differed depending on the clinical setting.

The study, provides empirical support for the applicability of the pace-of-life hypothesis [1] to hospitals. Pace-of-life can be measured in different ways to rank hospitals based on this construct, similar to previous work ranking countries by their pace-of-life [1]. Compared to Levine and Norenzayan's study [20], where it took citizens 12.17 to 18.33 seconds, hospital pace was towards the faster end of the city walkers (13–14 seconds to walk 20 metres in a public corridor). Our walking pace measurements did not show a difference between individual hospitals, possibly due to data collection limitations. The WOMBAT tool measured to the nearest second, but for this short pathway, this may not have been sufficiently accurate. Researchers at either end of the 20m walking strip signalled discretely to the timekeeper when the observed people crossed the start and finish line to prevent errors due to parallax, yet it was not as accurate as Levine and Norenzayan's stopwatch. Due to the minimal variation between sites, we did not use this measure in further analyses, but note that the data can still be used as a baseline measure or as a comparison for other hospitals or countries to compare hospital pace-of-life.

Our study provides empirical support to the Goldilocks hypothesis [4]. We found that staff working in acute care workplaces (i.e., emergency departments) rated their work higher on the pace scale compared to staff working in less-acute care workplaces (i.e., rehabilitation wards, aged care, and palliative care wards). This makes intuitive sense as a faster pace is related to distress and acuity of the patient, and time-critical and complex nature of tasks being performed. We found no significant relation to staff well-being as previously predicted [4]. While this could [35] be a result of the small comparative sample to conduct this analysis (n = 42 vs. n = 32), it may also indicate a more complex or nuanced understanding of pace.

The pace subscale from Ballard and Seibold (6) did not assign a positive or negative sentiment to the words used to describe the pace of work. While "hurried" or "racing" are most

likely perceived as negative, it is possible that "fast-paced" could be seen as a neutral or positive attribute. Saying your job is "fast-paced" may indicate an exciting, active and desirable place to work. Together, the items in the perceived pace scale showed high inter-item reliability suggesting "fast-paced" was seen as either negative or neutral. Our study also provided some qualitative evidence to support a faster pace of work in the hospital being a negative construct.

A rapid transactional pace is generally perceived by the person receiving the service as positive, so long as it is done with care and efficiency. This is the Goldilocks zone [4], that is, fast enough to be efficient but slow enough to be thorough and respectful. Our study shows that for those providing the service, a rapid pace has negative connotations, being correlated with lower job satisfaction and higher burnout.

## Conclusion

This paper is the first to investigate the relationship between pace, organizational culture, staff well-being and patient outcomes. It is also the first study to consider the socio-psychological theory of pace-of-life in the hospital context. Our paper adds to the knowledge base about the relationships and interactions between perceptions of pace, culture and staff and patient outcomes by applying some novel tools and methods to collect data. Having shown the feasibility and limitations of the methods, the next step is to refine these so that more robust testing of the Goldilocks effect is possible, i.e., what is the sweet spot for pace in hospitals?

## Supporting information

**S1 Appendix.**
(DOCX)

## Acknowledgments

The authors thank hospital staff participating in the survey.

## Author Contributions

**Conceptualization:** Janet C. Long, Louise A. Ellis, Kate Churruca, Jeffrey Braithwaite.

**Data curation:** Janet C. Long, Chiara Pomare, Louise A. Ellis, Kate Churruca.

**Formal analysis:** Janet C. Long, Chiara Pomare, Louise A. Ellis, Kate Churruca.

**Methodology:** Janet C. Long, Chiara Pomare, Louise A. Ellis, Kate Churruca, Jeffrey Braithwaite.

**Project administration:** Chiara Pomare.

**Validation:** Jeffrey Braithwaite.

**Writing – original draft:** Janet C. Long, Chiara Pomare, Louise A. Ellis, Kate Churruca.

**Writing – review & editing:** Janet C. Long, Chiara Pomare, Louise A. Ellis, Kate Churruca, Jeffrey Braithwaite.

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
