## [Decision Letter · Decision Letter 0]

4 Jun 2021

PONE-D-20-33878

The Pace of Hospital Life: A mixed methods study

PLOS ONE

Dear Dr. Long,

Thank you for submitting your manuscript to PLOS ONE. After careful consideration, we feel that it has merit but does not fully meet PLOS ONE’s publication criteria as it currently stands. Therefore, we invite you to submit a revised version of the manuscript that addresses the points raised during the review process.

The authors have assumed that all fast-pace hospital activity would not only lead to work dissatisfaction but also, as a consequence, would lead to less effective patient care. It is not a logical conclusion that fast-paced hospital activity would damage patient health care. This is because faster paced, and potentially more attentive patient care could very well lead to a higher standard of patient care and decreased patient illness and mortality. The authors do not mention this major counter hypothesis. But the broadest finding in the literature of hospital care is that more extensive and repetitive medical and surgical procedures are associated with beneficial patient outcomes. The authors should explore this counter hypothesis, both in terms of the actual intensiveness of work and the higher stress levels it tends to produce. Stress levels in the working staff could as easily indicate a high level of tension-related performance.

We look forward to receiving your revised manuscript.

Kind regards,

M. Harvey Brenner, PhD

Academic Editor

PLOS ONE

Journal Requirements:

2. Thank you for including your ethics statement:  "The ethical conduct of this study was approved by the appropriate Local Health District (HREC ref no: 16/363). Governance approvals to conduct the research were obtained for each hospital site.

We note that for the observation of walking speed, we used an “opt-out” process: “Walkers were blinded to the nature of the study and the purpose of the observation, but posters at the sites noted the presence of researchers and data collection activity. Staff members were able to opt-out (Hunt, Shlomo, & Addington-Hall, 2013) by discretely signalling “no” to the researchers.".   

a.) Please amend your current ethics statement to include the full name of the ethics committee/institutional review board(s) that approved your specific study.

b.) Please provide additional details regarding participant consent. In the ethics statement in the Methods and online submission information, please ensure that you have specified (1) whether consent was informed and (2) what type you obtained (for instance, written or verbal, and if verbal, how it was documented and witnessed). If your study included minors, state whether you obtained consent from parents or guardians. If the need for consent was waived by the ethics committee, please include this information.

3. Please include additional information regarding the survey or questionnaire used in the study and ensure that you have provided sufficient details that others could replicate the analyses. For instance, if you developed a questionnaire as part of this study and it is not under a copyright more restrictive than CC-BY, please include a copy, in both the original language and English, as Supporting Information, or include a citation if it has been published previously.

Reviewers' comments:

Reviewer's Responses to Questions

**Comments to the Author**

1. Is the manuscript technically sound, and do the data support the conclusions?

Reviewer #1: Yes

Reviewer #2: Yes

2. Has the statistical analysis been performed appropriately and rigorously? 

Reviewer #1: Yes

Reviewer #2: Yes

3. Have the authors made all data underlying the findings in their manuscript fully available?

Reviewer #1: No

Reviewer #2: Yes

4. Is the manuscript presented in an intelligible fashion and written in standard English?

Reviewer #1: Yes

Reviewer #2: Yes

5. Review Comments to the Author

Reviewer #1: At first I did not quite understand the purpose of the paper. At least it seemed somewhat obvious regarding the fast pace of hospitals. As I read further, it seems that the methods used were rigorous. Specifically the authors went beyond mere survey data and conducted direct observation. It would be good to make this clearer. The casual reader may not be aware that the methods went beyond just taking a survey.

It was unclear to me from the paper why people are rushing around so much in the hospital. Is there a reason for this - beyond the obvious need to keep pace with the workload? And does all this rushing around actually achieve anything? I would like to see some attempt to explain why some hospitals are working at more of a burnout pace. One metric that is often used are nurse-to-patient ratios. If hospitals do not have enough nurses - then they may have to rush around more. However, there must be a limit to this. The ability of one nurse to do the work of two nurses is finite. This is a huge problem in the US and has been made exceedingly worse by the current pandemic. There are just not enough nurses and in spite of their best efforts much work never gets done and patients die in part due to a shortage of nurses.

Reviewer #2: This paper, on the “Pace of Hospital Life” in Australian hospitals finds that a “fast-paced” “hurried” and “rapid” pace of life leads to negative perceptions of organizational culture, higher burn-out and lower job satisfaction. However, perceived pace did not predict patient safety.

The authors have assumed that all fast-pace hospital activity would not only lead to work dissatisfaction but also, as a consequence, would lead to less effective patient care. This did not occur.

On the other hand, it is not a logical conclusion that fast-paced hospital activity would damage patient health care. This is because faster paced, and potentially more attentive patient care could very well lead to a higher standard of patient care and decreased patient illness and mortality. The authors do not mention this major counter hypothesis. But the broadest finding in the literature of hospital care is that more extensive and repetitive medical and surgical procedures are associated with beneficial patient outcomes. The authors should explore this counter hypothesis, both in terms of the actual intensiveness of work and the higher stress levels it tends to produce. Stress levels in the working staff could as easily indicate a high level of tension-related performance.

6. PLOS authors have the option to publish the peer review history of their article (what does this mean?). If published, this will include your full peer review and any attached files.

Reviewer #1: No

Reviewer #2: **Yes: **M Harvey Brenner

---

## [Editor Report · Decision Letter 1]

26 Jul 2021

The Pace of Hospital Life: A mixed methods study

PONE-D-20-33878R1

Dear Dr. Long,

We’re pleased to inform you that your manuscript has been judged scientifically suitable for publication and will be formally accepted for publication once it meets all outstanding technical requirements.

Kind regards,

M. Harvey Brenner, PhD

Academic Editor

PLOS ONE
---

## [Editor Report · Acceptance letter]

10 Aug 2021

PONE-D-20-33878R1 

The Pace of Hospital Life:  A mixed methods study 

Dear Dr. Long:

I'm pleased to inform you that your manuscript has been deemed suitable for publication in PLOS ONE. Congratulations! Your manuscript is now with our production department. 

Kind regards, 

on behalf of

Professor M. Harvey Brenner 

Academic Editor

PLOS ONE